# Colombian Essential Oil of *Ruta graveolens* against Nosocomial Antifungal Resistant *Candida* Strains

**DOI:** 10.3390/jof7050383

**Published:** 2021-05-14

**Authors:** Matthew Gavino Donadu, Yeimmy Peralta-Ruiz, Donatella Usai, Francesca Maggio, Junior Bernando Molina-Hernandez, Davide Rizzo, Francesco Bussu, Salvatore Rubino, Stefania Zanetti, Antonello Paparella, Clemencia Chaves-Lopez

**Affiliations:** 1Department of Chemistry and Pharmacy, University of Sassari, 07100 Sassari, Italy; mdonadu@uniss.it; 2Department of Biomedical Sciences, University of Sassari, 07100 Sassari, Italy; srubino@uniss.it (S.R.); zanettis@uniss.it (S.Z.); 3Faculty of Bioscience and Technology for Food, Agriculture and Environment, University of Teramo, Via R. Balzarini 1, 64100 Teramo, Italy; yyperaltaruiz@unite.it (Y.P.-R.); jbmolinahernandez@unite.it (J.B.M.-H.); apaparella@unite.it (A.P.); 4Programa de Ingeniería Agroindustrial, Facultad de Ingeniería, Universidad del Atlántico, Carrera 30 Número 8-49, Puerto Colombia 081008, Colombia; dusai@uniss.it; 5Otolaryngology Division, Department of Medical, Surgical and Experimental Sciences, University of Sassari, 07100 Sassari, Italy; davide.rizzo@aousassari.it (D.R.); francesco.bussu.md@gmail.com (F.B.)

**Keywords:** *Candida* spp., innovative antifungals, head neck cancer, azole resistant, *Ruta graveolens*

## Abstract

Drug resistance in antifungal therapy, a problem unknown until a few years ago, is increasingly assuming importance especially in immunosuppressed patients and patients receiving chemotherapy and radiotherapy. In the past years, the use of essential oils as an approach to improve the effectiveness of antifungal agents and to reduce antifungal resistance levels has been proposed. Our research aimed to evaluate the antifungal activity of Colombian rue, *Ruta graveolens*, essential oil (REO) against clinical strains of *Candida albicans*, *Candida parapsilopsis*, *Candida glabrata*, and *Candida tropicalis*. Data obtained showed that *C. tropicalis* and *C. albicans* were the most sensitive strains showing minimum inhibitory concentrations (MIC) of 4.1 and 8.2 µg/mL of REO. Time–kill kinetics assay demonstrated that REO showed a fungicidal effect against *C. tropicalis* and a fungistatic effect against *C. albicans*. In addition, an amount of 40% of the biofilm formed by *C. albicans* was eradicated using 8.2 µg/mL of REO after 1 h of exposure. The synergistic effect of REO together with some antifungal compounds was also investigated. Fractional inhibitory concentration index (FICI) showed synergic effects of REO combined with amphotericin B. REO Lead a disruption in the cellular membrane integrity, consequently resulting in increased intracellular leakage of the macromolecules, thus confirming that the plasma membrane is a target of the mode of action of REO against *C. albicans* and *C. tropicalis.*

## 1. Introduction

The mucosal surfaces primarily affected by candidiasis are the oral cavity, esophagus, angles of the mouth, and genitals [1]. Oral candidiasis (OC) is a common fungal disease caused by *Candida* spp. with the appearance of white lesions generally affecting the oral or oropharyngeal mucosa [2]. Despite the progress in retroviral therapy, OC remains the most common cause of infections in immunocompromised patients affected by diseases such as the human immunodeficiency virus (HIV) [3]. Reports indicated that during the progression of their condition, more than 90% of people infected with HIV develop debilitating infections as oropharyngeal and esophageal candidiasis when they do not receive highly active antiretroviral therapy [4]. Furthermore, *Candida* species are the most common pathogen isolated in patients in the critical care setting. It is commonly found in elderly subjects, diabetic patients, and solid organ transplant recipients, and it is also an etiological agent of urinary and vaginal tract infections [4].

Although *C. albicans* is the predominant pathogenic fungus responsible for the OC [5], non-*albicans Candida* (NAC) species are starting to be frequently isolated in *Candida* infections. The incidence of species such as *C. glabrata*, *C. parapsilosis*, *C. tropicalis* has been widely reported within the past 10-year period [6]. In particular, *C. glabrata* and *C. parapsilosis* are frequently isolated in North and Central Europe and North America, and *C. tropicalis* in South America and Asia [7]. Moreover, the potential of these species to exhibit resistance and cross resistance to azole drugs, which might lead to the failure of therapeutic strategies, has been documented [8].

In order to find new classes of antifungals, the use of essential oils (EOs) has been proposed, and many studies have focused on studying EOs properties and application in fungal control [9,10]. Several works have reported the efficacy of EOs as a strategy against different preharvest and postharvest pathogens in fruits [11] and in human invasive fungal infection. In this regard, some studies reported the efficacy of *Satureja montana*; *Thymus capitatus*, and *Melaleuca alternifolia* EO in *Candida albicans* inhibition [12].

*Ruta graveolens* is a plant used in traditional and herbal medicine. It was used in some medieval rites to protect the house against negativity. In folk medicine, rue has been used to treat cough, diphtheria laryngitis, colic, headache, and as an antidote in case of mushroom poisoning, snake bites, and insect bites; in addition, it has been used for its stimulating, stomachic, emmenagogue effects consumed as an infusion and to treat headache, muscular and joint pain, as well as an anti-inflammatory using the oil or extract [13]. In the Middle Ages, in fact, rue was used to ward off the plague: its smell is in fact very strong and pungent. Nowadays, in some Latin American countries, it is used as fungicide and pesticide in organic agriculture [14,15]. In this regard, previous studies have demonstrated the effectiveness of *R. graveolens* essential oil (REO) in vitro against phytopathogens as *Colletotrichum gloeosporioides* [16,17], *Cladosporium herbarum*, *Aspergillus fumigatus*, *Fusarium oxysporum*, *Aspergillus flavus*, and *Alternaria alternata* [18]. Moreover, studies in situ on guava [17], papaya [16], gooseberries [19], tomato [20], and pear [21] showed a remarkable reduction of fruit decay and physicochemical properties preservation. Although data can be found in the scientific literature on REO effects against phytopathogens, a limited number of studies were reported on REO activity in human pathogens.

The present study aimed to clarify *R. graveolens* antifungal activity against multi-resistant *Candida* spp. of clinical origin at the same time evaluating the time–kill kinetics and the ability to reduce biofilm formation in order to find new alternatives to help overcome drug resistance in *Candida* spp.

## 2. Materials and Methods

### 2.1. Strains

A collection of 24 clinical isolates belonging to 4 different *Candida* spp. was selected for this study: *C. albicans* (6), *C. parapsilosis* (6), *C. tropicalis* (6), *C. glabrata* (6). The isolates were cultured from specimens isolated from the oral cavity of patients with head and neck cancer at the Otolaryngology Clinic, Department of Medical, Surgical and Experimental Sciences, University of Sassari, Italy. All microorganisms were identified by standard methods: germ tube test (GTT) and YBC Vitek Card (Bio-Merieux Marcy l’Etoile, France) [22] and stored on Sabouraud dextrose agar plates until the study was performed.

The identification of *C. tropicalis* ORL20 and ORL21 was successively confirmed using the amplification of the ITS region (ITS1-5, 8S-ITS2) with universal fungal primers (ITS1, ITS4) [23]. The GenBank accession number of the ITS sequences obtained are *C. tropicalis* ORL20: KX664640.1 and *C. tropicalis* ORL21 KX664611.1.

### 2.2. Reagents

Fluconazole (FLC) was obtained from Sigma-Aldrich. Stock solutions of FLC were prepared in dimethyl sulfoxide. The final concentration of DMSO was not higher than 0.14%. In addition, RPMI 1640 (Thermo Fisher Scientific) was used in this study. Rue essential oil (REO) was obtained from Kräuter SAS (Bogotá, Colombia) lot n ° SSTE01.

### 2.3. Antifungal Susceptibility Testing

The minimum inhibitory concentrations (MICs) of antifungal agents (REO and FLC) against the *Candida* spp. strains were determined according to the broth microdilution assay in 96-well microtitration plates, as described by the M27-A3 method from the Clinical and Laboratory Standards Institute (CLSI, formerly NCCLS) [24]. Twofold serial dilutions in RPMI 1640 medium were performed to obtain the final concentrations that ranged from 0.029 to 131 µg/mL for REO and from 0.125 to 512 µg/mL for FLC. The test was carried out in a final volume of 200 µL total per well as follows: 100 µL of the culture medium and 100 µL of fungal inoculum at a concentration of 10^6^ CFU/mL. Each strain was tested in duplicate and positive growth control (the strain under test without REO) and a negative one (medium only) were included in each test. The plate was incubated at 37 °C, and the minimal fungicide concentration (MFC) was determined by taking 10 µL from each well and spreading them on Sabouraud dextrose agar. The plates were incubated at 37 °C for 24/48 h and checked to detect microbial growth. MFC is considered the lowest concentration capable of inhibiting 99% fungal growth. Three independent experiments were performed.

### 2.4. Determination of Minimum Fungicidal Concentration (MFC)

In order to establish the MFC of *Candida* species, the broth dilution method was used, as recommended by the Clinical and Laboratory Standard Institute (CLSI. 2008). Yeasts were cultivated at 37 °C on Sabouraud dextrose agar plates (Microbiol, Cagliari, Italy) for 24 h. The inoculum was prepared by a dilution of the colonies in a salt solution, at a concentration of 0.5 McFarland and confirming the concentration by spectrophotometric reading at a wavelength of 530 nm. The sensitivity test was carried out in RPMI 1640, using 96-well plates. Oil concentrations were prepared by serial one to two dilutions from 131 to 1.0 µg/mL. After shaking, 100 µL of each oil dilution and 100 µL of yeast suspension at a concentration of 10^6^ CFU/mL were added to each well and then incubated at 37 °C for 48 h.

In order to determine the MFC value, 10 µL were seeded on Sabouraud dextrose medium, the plates were incubated for 24–48 h at the temperature of 37 °C. Minimal fungicidal concentration (MFC) was considered as the lowest concentration inhibiting fungal growth. Moreover, each yeast strain included in the study was tested for its sensitivity to fluconazole, voriconazole, and amphotericin B. Each experiment was performed in duplicate and repeated three times.

Two *C. albicans* (ORL3 and ORL8) and two *C. tropicalis* (ORL20 and ORL21) strains that were selected for their sensibility to REO and amphotericin B, were then chosen for further studies.

### 2.5. Synergistic Potential of REO with Antifungal Antibiotics against C. albicans and C. tropicalis

In order to determine the synergy between antifungal antibiotics and *Ruta graveolens* essential oil, the checkerboard method was performed to obtain the Fractional inhibitory concentration indices (FICIs) of REO in combination with amphotericin B and fluconazole, following the methodology proposed by [25]. The microtiter plates were filled with a combination of 50 µL REO and 50 µL antibiotics (fluconazole and amphotericin B) at MIC diluted twofold in a serial manner in YPD broth with a 0.01% of 2,3,5-Triphenyltetrazolium chloride (TPC). Successively, 10 µL of fungal culture were seeded (10^6^ CFU/mL) in each well and mixed well. The plates were incubated at 37 °C for 48 h. The inhibition of the growth of fungal cells was indicated by the absence of the red color.

The FICIs were calculated using the following formulas:$$FICI=FIC_{REO}+FIC_{antibiotics}$$
$$FIC_{REO}=\frac{MIC\;of\;REO\;in\;combination}{MIC\;of\;REO\;alone}$$
$$FIC_{Antibiotic}=\frac{MIC\;of\;Antibiotic\;\;in\;combination}{MIC\;of\;Antibiotic\;alone}$$

The interpretation of FIC indices (FICIs) was made following the approach used by Fratini et al. [26]. This method could give a more precise interpretation of FICs since it overcomes the MIC quantification error when synergy is determined by the method reported by Odds et al. [27]. The FICs were considered as synergistic effect (FIC Index ≤ 1.0); commutative effect (FIC Index = 1); no interaction (1.0 <FIC Index ≤2.0); and antagonistic effect (FIC Index > 2.0).

### 2.6. Time–Kill Kinetics

In order to further evaluate the REO effect, the time–kill assay in two *C. albicans* and two *C. tropicalis* strains was performed following the method proposed by Chaves-López et al. [28], with some modifications. A cell suspension was prepared for each strain, starting with an inoculum of 4.5 ± 0.5 and 5.0 ± 0.5 log CFU/mL and inoculating it into an emulsion of 8.2 µg/mL of REO in yeast potato dextrose broth (YPD). The suspension was incubated at 37 °C for 48 h, and an aliquot of 1 mL was taken in the times 0, 1, 2, 3, 4, 5, 24, 36, and 48 h to prepare a series of 10-fold dilutions to subsequently inoculate 0.1 mL in Petri dishes with YPD agar. After incubation at 37 °C for 48 h, the colonies were counted for each dilution. All microbiological tests were repeated in two different experiments. Each experiment was performed in triplicate.

In order to determine REO fungicidal or fungistatic features, a reduction of <3 log CFU/mL after the treatment in the growth of the starting inoculum was defined as fungistatic activity of REO, and a reduction ≥3 log CFU/mL as fungicidal activity according to the method proposed by Scorneaux et al. [29]. Subsequently, the time necessary to achieve a 50, 90, and 99% of reduction in growth from the starting inoculum was determined.

### 2.7. Quantitative Assessment of Biofilm Formation

Standardized samples from four *Candida* strains were evaluated to quantitate the reduction of biofilm in the presence of REO in 96-well polystyrene microplates according to the methodology reported by Rossi et al. and Chaves-López et al. [30,31]. To biofilm formation, 200 µL of the sample was cultured in each well in YPD broth and incubated at 37 °C for 48 h. Then, the YPD broth was removed from the microplate, and 200 µL 8.2 µg/mL REO–YPD emulsion was added, with incubation at 37 °C for one hour. Then, the floating cells were removed, and the biofilm at the bottom of the wells was washed with deionized water three times. Six replicates were dispensed of each sample, and cultures without REO were taken as control, and as a positive control was used amphotericin B, a well-known compound with an antibiofilm agent. The reduction of biofilm was quantified by staining the wells with 0.1% crystal violet (Sigma-Aldrich, Italy) for 20 min at room temperature. Samples were rewashed with deionized water until the removal of the excess dye. Finally, the samples were soaked in 250 μL of 30% glacial acetic acid (Carlo Erba Reagents, Italy).

The absorbances values at 590 nm (OD_590_) for each strain were measured using a Biolog MicroStation system (Biolog Inc., Hayward, USA), and the biofilm productions were grouped into: OD_590_ < 0.1 = nonproducers (NP), OD_590_ 0.1–1.0 = weak producers (WP), OD_590_ 1.1–3.0 = moderate producers (MP), and OD_590_ > 3.0 = strong producers (SP). The biofilm reduction was calculated using the following equation:$$\%\;Biofilm\;reduction=\frac{Abs_{CO}-Abs_{REO}}{Abs_{CO}}\times 100$$
where Abs_CO_ = absorbance sample control and Abs_REO_ = absorbance sample treated with REO.

### 2.8. Leakage of DNA and RNA through the Fungal Membrane

The release of cellular contents was determined according to the method by [32], with some modifications; we used a yeast load 5.0 ± 0.5 log CFU/mL for the analysis. To determine the concentration of the released constituents at 0, 30, 60, and 120 min of treatment, 50 μL of supernatant after centrifugation was used to measure the absorbance at 260 nm with a bio photometer (Eppendorf 6131, Eppendorf, Hamburg, Germany). Control samples without REO, with REO, and with fluconazole were tested.

### 2.9. Measurement of Extracellular pH

*Candida* extracellular pH treated with 8.2 µg/mL of REO was determined according to the methodology reported by [32]. We used a yeast load 5.0 ± 0.5 log CFU/mL for the analysis. After the centrifugation of the cells, these were washed three times and resuspended with sterilized double-distilled water. The measurements of extracellular pH of the samples at 0, 30, 60, and 120 min were carried out using a pH-meter (Mettler-Toledo Greisensee, Switzerland), and a control sample without REO, and with fluconazole were tested.

### 2.10. Effect of the REO on the Membrane Integrity

Cell membrane damage produced by the REO was evidenced using Evans blue staining according to Chaves-Lopez et al. [33]. Briefly, 20 µL of the *Candida* suspension were incubated in coverslips in YPD broth at 37 °C for 24 h. Then, one of them for each strain was treated with 8.2 µg/mL of REO for one hour and they were stained with Evans blue staining for five minutes. No treated samples were considered as control. Samples were observed under microscopy (Nikon ECLIPSE E 200, Nikon, Melville, NY, USA). Photographs were taken with Samsung COLOR CAMERA SAC-410 PA interfaced with a PC.

### 2.11. Data Analysis

Experimental results were expressed as means ± standard deviations: data were evaluated by analysis of variance (ANOVA) and compared by 95% Tukey’s HSD test, using Statistica 13.5 software (TIBCO, Tulsa, OK, USA).

## 3. Results

### 3.1. Oil Characterization

Data regarding the REO characterization by mass spectrometry–gas chromatography (MS–GC) were reported in our previous work [17] (Appendix A). REO showed a predominant content of aliphatic ketones in which 2-undecanone was the major component in the oil.

### 3.2. Antifungal Susceptibility Testing

The antifungal activities of REO and FLC alone were determined by broth microdilution assay. Among the 24 isolates of *Candida* species tested, 8 isolates were resistant to FLC with MIC values ranging from 8.2 to 256 µg/mL, and 8 isolates were sensitive to FLC with MICs ranging from 1 to 4 µg/mL. The MICs of REO were in a range of 8.2–131 µg/mL against all the *Candida* spp. isolates (Table 1).

### 3.3. Determination of Minimum Fungicidal Concentration (MFC)

The MFC data for clinical *Candida* species in Table 2 show that after 24 h the values obtained with REO essential oil were 8.2 µg/mL, 131–66 µg/mL, 4.2 µg/mL, 16.4 µg/mL, and 4.1–16.4 µg/mL for *C. albicans*, *C. glabrata*, *C. tropicalis*, and *C. parapsilosis*, respectively. After 48 h, the MFCs obtained were 8.2–12.3 µg/mL for *C. albicans*, 131 µg/mL for *C. glabrata*, 8.2 µg/mL for *C. tropicalis*, and 16.4 µg/mL for *C. parapsilosis*. Furthermore, *C. glabrata* and *C. parapsilosis* were resistant to fluconazole (MFC: 128 and 256 µg/mL after 24 h and 2 and 4 µg/mL after 48 h, respectively); in addition, *C. glabrata* was also resistant to voriconazole (MFC: 2 µg/mL after 24 h and 4 µg/mL after 48 h). The antifungal effect of REO was therefore highlighted against *C. albicans* and *C. tropicalis* also with respect to synthetic drugs such as amphotericin B and fluconazole.

It should be pointed out that, generally, the efficiency of REO for the yeast species here studied is quite different not only for different species but even among the same species.

### 3.4. Synergistic Activity of REO with Antifungal Antibiotics

To overcome the mechanisms of bacterial resistance against antibiotics, in the last years, some studies have proposed the use of association of plant extracts with antibiotics. In our study, the synergistic potential of essential oil of *Ruta graveolens* in combination with fluconazole and amphotericin B using the checkerboard method was evaluated. As shown in Table 3, REO showed synergistic effects with amphotericin B against *C. albicans* ORL3 and ORL8 and *C. tropicalis* ORL21 with FICI values of 0.38, 0.5, and 0.8, respectively. REO did not present interaction with the amphotericin B in *C. tropicalis* ORL20. Regarding the combination of REO–fluconazole, no interactions were evidenced with *C. tropicalis* or *C. albicans* ORL8. However, antagonistic activity was revealed in *C. albicans* ORL3. As is observed in Table 3, the concentrations used to achieve the synergistic activity of the combinations were considerably lower than those of the MIC of oil and antibiotics used and despite in some combinations no evidenced synergy if a decrease in the individual concentration used in the drugs was reached.

### 3.5. Time–Kill Kinetics (TKK)

For the TKK assay, two representative strains of *C. albicans* and *C. tropicalis* that showed high sensibility to REO were taken into consideration.

REO and fluconazole mean time–kill graphs and standard deviations against the four *Candida* strains tested are depicted in Figure 1. The kinetics of inactivation monitored over 48 h evidenced the activity of REO against the strains tested with a fungicidal and fungistatic behavior, confirming that the *C. tropicalis* strains were more sensitive to the treatment than *C. albicans*. In addition, differences between the strains of the same species were also observed. Then, 15 min after the treatment, indeed, the cell count was reduced by about 1.4 log CFU/mL. and 1.0 for C. *tropicalis* ORL21 and *C. tropicalis* ORL20, respectively. After 2 h of treatment, there was a further decrease of about 1.7–1.5 log CFU/mL for both strains. With exposure time increase, both strains were reduced, further reaching values of 1.5 log CFU/mL and 1.15 log CFU/mL, respectively, thus evidencing a fungicidal activity (a kill of ≥3 log CFU/mL).

The effect of REO was more reduced in *C. albicans*, showing only 0.98 log CFU/mL after 2 h of exposure in *C. albicans* ORL08 and 0.25 log CFU/mL in *C. albicans* ORL03. Additionally, in this case, there was a further reduction of the yeast population achieving counts of 3.58 and 2.92 log CFU/mL for *Candida albicans* ORL08 and ORL03, respectively, reflecting a fungistatic activity. Additionally, it was found that concurrent time–kill experiments on isolates with fluconazole failed to show reductions in starting inoculum.

The time required by REO to achieve a reduction of growth of the starting inoculum was determined for each strain (Table 4). For *C. tropicalis*, the time needed to reach 50% of the reduction was less than an hour but it presented differences between ORL20 and ORL21 strains, being the latter the most sensitive to REO with 0.39 h. After about 1.5 h. a 90% reduction of growth was evidenced by both strains, reaching 99.9% at 1.8 and 2.9 h for *C. tropicalis* ORL21 and ORL20, respectively. Concerning *C. albicans*, the reduction of 50% of growth with respect to the starting inoculum was reached in a range of 3.6 and 4.5 h. No was achieved a reduction in the CFU of 90% and 99.9%.

### 3.6. Biofilm Reduction

All the strains tested produced significant biofilm biomass on polystyrene microplates at 37 °C, as shown in Figure 2. The two *C. tropicalis* and *C. albicans* ORL3 strains formed strong biofilm biomass, while *C. albicans* ORL08 showed a weak production. When 8.2 µg/mL of REO was applied to the formed biofilm, we observed a significant detachment of the biofilm biomass after 1 h of REO exposure, overall, in *C. albicans* (41.2 and 36.1% for ORL08 and ORL03 strains, respectively). On the contrary, we observed a negligible biofilm detachment in samples with *C. tropicalis* during the first hour of treatment.

Comparing the REO efficacy with that of amphotericin B, on the biofilm eradication, we observed slight but not significant (*p* > 0.05) differences between the two treatments. It is worth mentioning that the strongest biofilm formed by *C. albicans* ORL03 was more difficult to eradicate with both REO and amphotericin B.

### 3.7. Leakage of DNA and RNA through the Fungal Membrane

The mechanism of the action of the antifungal activity of essential oils is not clear; some authors suggest that causes significant membrane damage due to the destruction of the membrane integrity [34]. The effect of REO on *Candida* membrane degradation and the release of cellular constituents at 0, 30, 60, and 120 min after the treatment were determined, and the results are shown in Figure 3.

As observed in Figure 3, the exposure of *C. tropicalis* and *C. albicans* to REO leads to a significant (*p* < 0.05) increase in the cellular release, which was intensified with the exposure time. Indeed, an early release of the intracellular compounds was observed already after 30 min of exposure to REO, in comparison with the untreated sample. A minimal and constant cellular release in the *Candida* yeasts was observed with the fluconazole.

### 3.8. Extracellular pH

The extracellular pH of *Candida* cells exposed to REO, fluconazole, and untreated are presented in Figure 4. The extracellular pH decreased in the control. Regarding the extracellular pH of the *C. Tropicalis*, yeasts treated with REO showed a constant behavior in the initial 30 min after treatment, followed by a significant increase (*p* < 0.05) of 15.2 and 14.2% after 120 min treatment for ORL21 and ORL20, respectively, concerning the control. *C. albicans* yeasts showed a minor but significant increase (*p* < 0.05) about control with 3.7 for ORL08, and 7.6% for ORL03. No significant differences were evidenced in the yeasts with the fluconazole after 120 min post treatment.

### 3.9. Cell Membrane Integrity

To investigate if there was a disruption of cell membrane integrity to the exposure to the REO, the cells were stained with Evans blue staining. As indicated in Figure 5, the results showed that when the yeasts are treated with the REO and observed under a light microscope the majority of the cells were blue stained, suggesting that the cell membranes were compromised after 1 h of treatment with the essential oil. Therefore, REO may act on the cellular membrane affecting its integrity, and consequently resulting in increased intracellular leakage of the macromolecules, thus confirming that the plasma membrane is a target of the mode of action of REO against *C. albicans* and *C. tropicalis.*

In addition, we observed a cell shrinkage after the treatment with REO probably due to the release of the intracellular components.

## 4. Discussion

Resistance may be due to an altered intracellular accumulation of the drug, an altered composition of membrane sterols, an alteration of ERG11 (the gene that encodes the enzyme lanosterol-14α-demethylase, the target of these drugs), or an alteration of the functionality of the efflux pumps [31]. These last two mechanisms are the most frequently called into question. The alteration of the target enzyme can be linked both to an upregulation of the gene that encodes it and to mutations of the gene itself. In the first case, the need is created for a higher intracellular concentration of azoles to be able to complex all the enzymatic molecules present in the cell, while in the second case, there is the production of a modified enzyme for which the drug has a reduced affinity. These mechanisms have been described in isolates of *C. albicans*, *C. neoformans*, and *Malassezia* sp. [35,36]. The intrinsic resistance to azole antifungals in *C. albicans* seems to be due precisely to a reduced susceptibility of the target enzyme. The other mechanism of resistance to azoles may occur due to the inability of antifungal agents to accumulate in the cell due to a high outflow of the drug in turn due to an alteration of the functionality of transporters located on the membrane of the fungus. Two types of transporters mediate this mechanism: the “ABC transporters”, encoded by the CDR genes, and the major facilitators, encoded by the MDR genes. These mechanisms have been described in isolates of *C. albicans*, *C. glabrata*, *C. parapsilosis*, and *C. tropicalis* [37,38].

In this study, we demonstrated that *Ruta graveolens* essential oil had a satisfactory antifungal activity against *C. tropicalis and C. albicans* associated with oral candidiasis. Preliminary studies demonstrated that the antifungal activity of this oil is due overall to the main components 2-nonanol and 2-undecanone, which exhibited the most potent antifungal effect [16]. On the other hand, Reddy et al. [39] tested REO against six diverse fungi species showing the most significant antifungal activity against *C. albicans*, with 86% of growth reduction in comparison to the positive control (amphotericin B); these authors showed that the antifungal activity is related to the abundance of ketones and alcohols in the REO. In addition, Attia et al. [40] also found antifungal activity against two *C. albicans* clinical strains, a *C. glabrata* with MIC of 1.14–2.5 µg/mL; moreover, morphological changes were observed including cell surface deformation, disruption, and prevention of germ tube production; additionally, it was demonstrated a direct correlation between the percentage of ketones and the antimicrobial activity.

Antibiotic resistance is a big concern around the world and many strategies have been adopted to reduce this problem. In the last years, a rational approach to deal with antibiotic resistance problems using a combination therapy combining conventional antibiotics and essential oils has been proposed. In this work, we evaluated the effect of these combinations using the checkboard method and the determination of the FICs was made using the novel approach proposed by Frattini et al. [26]. This approach reduces the errors caused by usual uncertainty in the determination of MIC value (mode ± 1 dilution), which can lead to numerous possibilities for reproducibility errors in the MIC checkboard, as reported by Odds et al. [27]. Using this approach, the synergistic effect detected in our experiment appears to be strain dependent, thus suggesting a possible presence in yeast, of some intrinsic factors that might potentiate the antifungal activity of the amphotericin B and REO when acting in tandem.

In this context, the interaction between plant extracts and antibiotics with synergistic activity against *Candida* species has been reported. For example, Saad et al. [41] reported that geraniol displayed a synergistic effect with both fluconazole and amphotericin B. Additionally, the synergistic effect of thymol and nystatin was observed on *Candida* species [42]. In the same way, *Citrus aurantium* essential oil showed synergistic potential with fluconazole and amphotericin B against *C. albicans* and enhanced the antifungal efficacy of the clinical drugs by 8.3 to 34.4 folds [25]. In this study, we reported for the first time a synergistic effect of REO with amphotericin B against *C. albicans* and *C. tropicalis*. Combination of clinical antibiotics with essential oils and phytocompounds targeting resistant fungi may have a different mechanism of action including (i) sequential inhibition of common biochemical pathways, (ii) amplified diffusion of one antifungal agent subsequent from the action of another antifungal agent on the fungal cell membrane, (iii) inhibition of different targets, and (iv) the inhibition of carrier proteins [42].

In our study, the time–kill kinetics demonstrated that REO had an effect against the four strains of *C. albicans* and *C. tropicalis* tested. REO evidenced the highest levels of killing against *C. tropicalis* strains showing antifungal activity. The time to achieve a 50% reduction of starting inoculum growth was less than an hour with a decrease in growth concerning control of 76.1% for *C. tropicalis ORL21* and 82% for *C. tropicalis ORL20* after 48 h of treatment. However, the REO effect was not so fast with *C. albicans* strains with just a 50% reduction after 3.5 h, a final reduction of 48% for *C. albicans ORL08*, and 50% for *C. albicans ORL03*, indicating fungistatic activity of the REO. These results suggest that REO effects depend on the *Candida* species. Similar results have been reported with other essential oils showing different fungicidal or fungistatic activity according to *Candida* species. In one study, *Ocimum gratissimum* L. essential oil was fungicide against *C. tropicalis* but showed fungistatic activity against *C. albicans* [43]. Some authors reported that the fungicidal effect of REO treated with salicylic acid in *C. glabrata* and *C. albicans* takes a time of over 1.5 h to reach a 50% of reduction of growth [40].

*Candida* biofilm is a well-organized formed by planktonic and mycelial yeast form, surrounded by extracellular polymeric substances; this structure is an effective microbial protection and can generate the well-known drug resistance [44]. REO demonstrated an antibiofilm action on the two *C. albicans* strains evaluated; this activity was detected within just one hour of treatment. The percentage of biofilm eradication here obtained were similar to those obtained with amphotericin B, a drug with known efficacy against *Candida* biofilm [45]. Studies reported an antibiofilm *Candida* activity of different EOs such as peppermint, eucalyptus, ginger grass, clove, and thyme essential oils ranging between 28 and 85% [43,44]. Some authors suggested that the antibiofilm capacity of EOs is related to the inhibition of filamentation and germ tube formation and interference with the cell membrane of planktonic and sessile cells of *Candida albicans*; furthermore, the hydrophobic character of EOs may increase the absorption through charged extracellular polymers, producing that oil has greater contact and permeation in the membrane to the cells [46,47]. REO has demonstrated a similar action mechanism in *C. gloesporioides*, where a compromised membrane was observed after one hour of exposure [16]. No significant biofilm eradication activity was observed with REO in *C. tropicalis* strains. Al-Fattani et al. [48] reported that biofilms of *C. tropicalis* are mainly constituted by hexosamine matrix in comparison with a glucose-rich matrix of the *C. albicans* biofilm; such a structure in the latter allows faster drug penetration. Moreover, one study involving several candida species aiming at measuring drug diffusion rates reported that the slowest rates of penetrations were presented by *C. tropicalis* [49]. Considering the above results, further studies will be addressed to increase the concentration and exposure time of REO to obtain the eradication of the *C. tropicalis* biofilm.

During recent years, the antifungal activity of the REO has been reported [16,17,18,19,20,21]; however, their mode of action is not yet clearly understood. Thus, in the present study, we used different approaches to provide insights into the antifungal activity of REO. For this purpose, we evaluated the release cellular, the extracellular pH, and the cell membrane integrity in *Candida*. Some authors have suggested that the antimicrobial activity of essential oils involves phenomena such as changes of cell membrane integrity, leading to an alteration of permeability and consequent leakage of cell contents [50]. In our experiment we observed a marked release of cell constituents during the time with a notable cell staining with the Evans blue staining, indicating that the permeability of the cell membrane was compromised. In addition, REO clearly induced the leakage of intracellular protons as evidenced by the increase in extracellular pH. Our results suggest that the REO accumulation in the cell membrane can induce loss of integrity changing the homeostatic environment, allowing ions leakage and cellular release, which could be responsible for the antifungal activity of this essential oil. No significant cellular release or changes in extracellular pH were presented with fluconazole evidencing the different mechanism of action that presents the azoles [51]. Some authors have reported similar results with other essential oils; Chen et al. [50] reported that *Anethum graveolens* essential oil induced a lesion of the cell membrane in *C. albicans.* On other hand, Ahmad et al. [52] reported that the mechanism of action of the *Coriaria nepalensis* essential oils disrupt membrane integrity in different *Candida* isolates. A similar result was reported by Rajkowska et al. [53] using different essential oils against *Candida*.

## 5. Conclusions

In this work, we explored the use of *Ruta graveolens* essential oils as a natural fungicidal agent and as a biofilm eradication agent and gave insights into its mode of action on *Candida albicans* and *Candida tropicalis*. Our study showed strong antifungal and biofilm eradication activity using 8.2 µg/mL of REO. In addition, we observed irreversible cellular membrane damage inducing leakage of the intracellular compounds very few times after the treatment. Fractional inhibitory concentration index (FICI) showed synergic effects of REO combined with amphotericin B. These results suggest the possible effective use of REO alone or in combination with amphotericin B, against multidrug resistant overall on *C. tropicalis* strains. In addition, the present study demonstrated that REO essential oil is a promising alternative for the treatment of *Candida* biofilms eradication.

## Figures and Tables

**Figure 1 jof-07-00383-f001:**
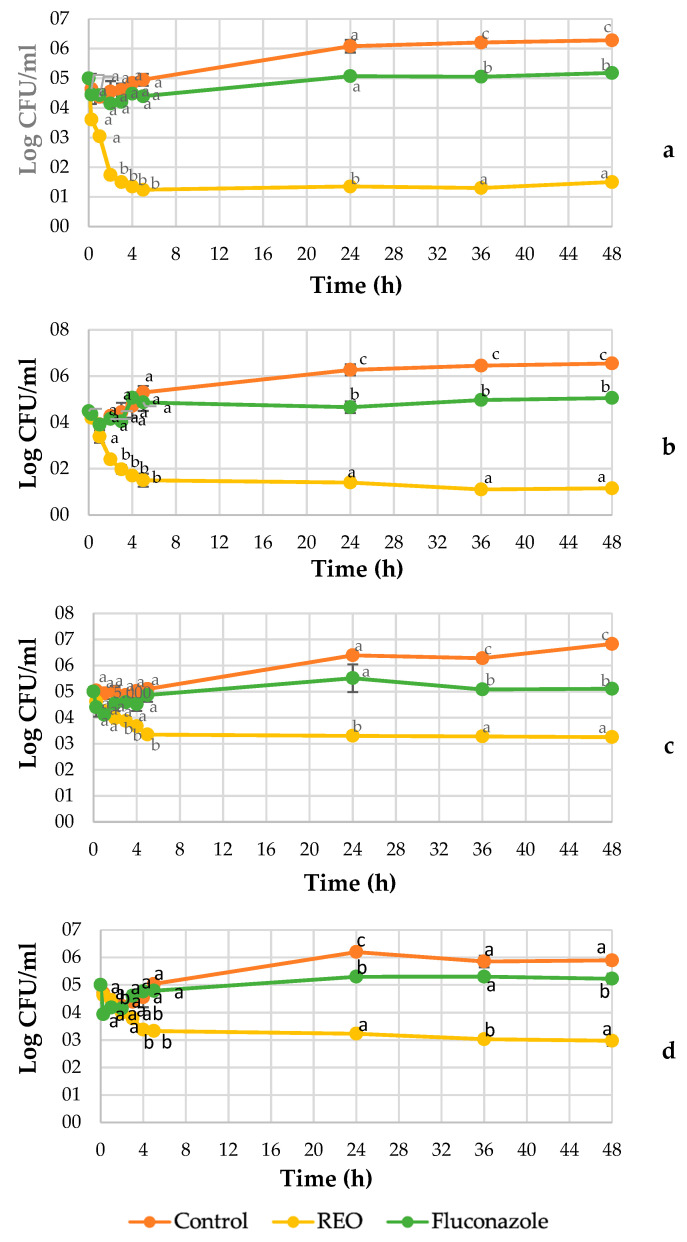
Time–kill kinetics for 8.2 µg/mL of REO and fluconazole against *Candida tropicalis* ORL21 (**a**), *Candida tropicalis* ORL20 (**b**), *Candida albicans* ORL08 (**c**), and *Candida albicans* ORL03 (**d**) at 37 °C. Results are according to ANOVA test mean values and intervals based on Tukey test for treatments with 95%. Different lowercase letters (a, b, c) indicate significant differences between treatments according to the Tukey test in a confidence interval of 95%.

**Figure 2 jof-07-00383-f002:**
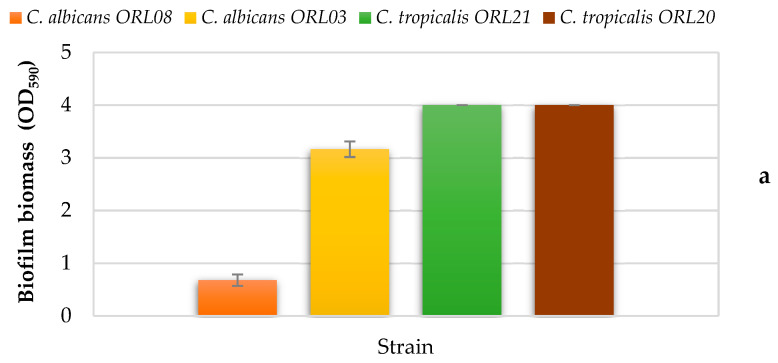
Effect of REO on biofilm of *C. tropicalis* and *C. albicans.* Formation of biofilm at 37 °C for 48 h, where OD_590_ < 0.1 = nonproducers (NP), OD_590_ 0.1–1.0 = weak producers (WP), OD_590_ 1.1–3.0 = moderate producers (MP), and OD_590_ > 3.0 = strong producers (SP) (**a**), percentage of eradication of biofilm after 1 h of REO treatment (8.2 µg/mL), amphotericin B (0.5 µg/mL) for *C. albicans* strains, and amphotericin B (1.0 µg/mL) for *C. tropicalis* strains (**b**). Results are according to ANOVA test mean values and intervals based on Tukey test for treatments with 95%. Different lowercase letters (a, b, c) indicate significant differences between treatments according to the Tukey test in a confidence interval of 95%. Different uppercase letters (A, B, C) indicate significant differences between strains according to the Tukey test in a confidence interval of 95%.

**Figure 3 jof-07-00383-f003:**
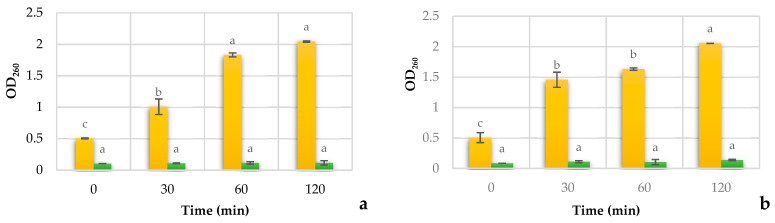
Effect of REO and Fluconazole on the 260 nm absorbing cell constituents release with the time. Release cell concentration of *C. Tropicalis* ORL21 (**a**), of *C. Tropicalis* ORL20 (**b**), of *C. albicans* ORL08 (**c**), and of *C. albicans* ORL03 (**d**) with the treatment at different time. The results are expressed as the absorbance of the sample (treated)–the absorbance of the control (no treated). Results are according to ANOVA test mean values and intervals based on Tukey test for time with 95%. Different lowercase letters (a, b, c) indicate significant differences between times according to the Tukey test in a confidence interval of 95%.

**Figure 4 jof-07-00383-f004:**
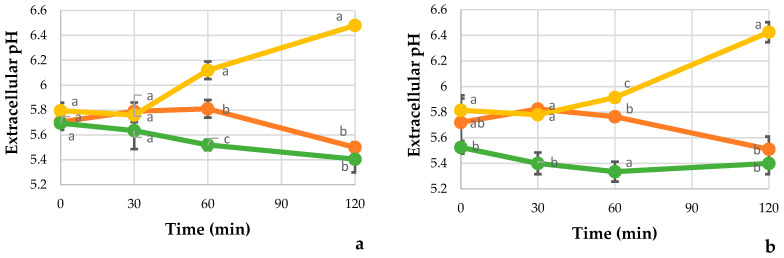
Extracellular pH of *Candida* yeast treated with REO and fluconazole. *C. tropicalis* ORL21 (**a**), *C. tropicalis* ORL20 (**b**), *C. albicans* ORL08 (**c**), and *C. albicans* ORL03 (**d**) with the treatment at different times. Values are the averages of the replicates for all the analyses. Error bars are ± SD of the means. Results are according to ANOVA test mean values and intervals based on Tukey test for treatments with 95%. Different lowercase letters (a, b, c) indicate significant differences between treatments according to the Tukey test in a confidence interval of 95%.

**Figure 5 jof-07-00383-f005:**
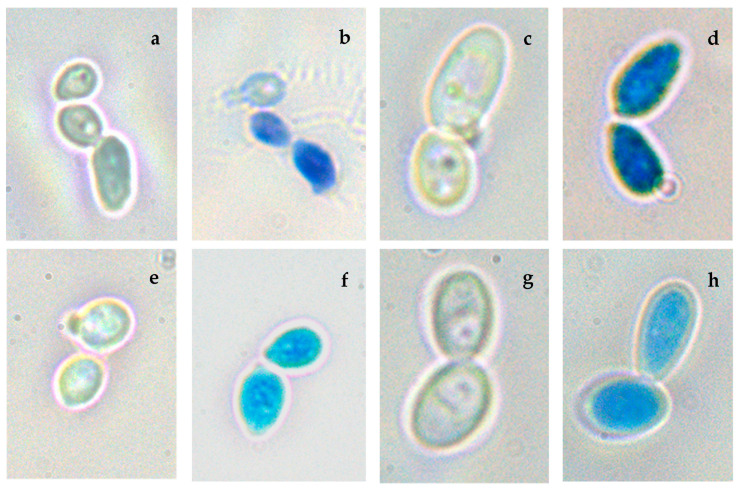
Microscopic observation (100×), of *C. tropicalis* (ORL21 and ORL20) and *C. albicans* (ORL08 and ORL3) before and after treatment with REO using Evans blue staining: (**a**,**c**,**e**,**g**) untreated controls and (**b**,**d**,**f**,**h**) cells treated with REO (8.2 µg/mL).

**Table 1 jof-07-00383-t001:** Inhibition effect of *Ruta graveolens* L. against different *Candida* species.

Fungi (n)	*Ruta graveolens* L.	Fluconazole
	Range (µg/mL)	Range
	MIC	(µg/mL)
*C. albicans* (6)	8.2 ± 0.5	0.25–1
*C. parapsilosis* (6)	16.4 ± 0.5	0.5–2
*C. tropicalis* (6)	4.1 ± 0.25	0.5–2
*C. glabrata* (6)	131 ± 0.5	8–256

**Table 2 jof-07-00383-t002:** *In vitro* susceptibility of *Candida* spp. isolates to *Ruta graveolens* essential oil and antifungal drugs.

Strains	*Ruta graveolens (REO)* (µg/mL) MFC		Amphotericin B (µg/mL) MFC		Fluconazole (µg/mL) MFC		Voriconazole (µg/mL) MFC	
	24 h	48 h	24 h	48 h	24 h	48 h	24 h	48 h
*C. albicans* ORL02	8.2 ± 0.5	12.3 ± 0.5	0.5 ± 0.25	0.5 ± 0.25	0.5 ± 0.25	1 ± 0.5	0.03 ± 0.005	0.03 ± 0.005
*C. albicans* ORL03	8.2 ± 0.5	8.2 ± 0.5	1 ± 0.5	0.5 ± 0.25	0.5 ± 0.25	1 ± 0.5	0.03 ± 0.005	0.03 ± 0.005
*C. albicans* ORL05	8.2 ± 0.5	12.3 ± 0.5	0.5 ± 0.25	1 ± 0.5	0.5 ± 0.25	1 ± 0.5	0.03 ± 0.005	0.03 ± 0.005
*C. albicans* ORL07	8.2 ± 0.5	16.4 ± 0.5	1 ± 0.5	1.5 ± 0.5	0.5 ± 0.25	1 ± 0.5	0.03 ± 0.005	0.03 ± 0.005
*C. albicans* ORL08	8.2 ± 0.5	8.2 ± 0.5	0.5 ± 0.25	0.5 ± 0.25	1 ± 0.5	1.5 ± 0.5	0.03 ± 0.005	0.03 ± 0.005
*C. albicans* ORL09	8.2 ± 0.5	8.2 ± 0.5	0.5 ± 0.25	0.5 ± 0.25	1 ± 0.5	1.5 ± 0.5	0.03 ± 0.005	0.03 ± 0.005
*C. glabrata* ORL02	131 ± 1	131 ± 1	2 ± 0.5	1 ± 0.5	128 ± 2	128 ± 2	2 ± 0.5	4 ± 0.5
*C. glabrata* ORL11	131 ± 1	131 ± 1	2 ± 0.5	1 ± 0.5	128 ± 2	256 ± 2	2 ± 0.5	4 ± 0.5
*C. glabrata* ORL15	66 ± 1	131 ± 1	1 ± 0.5	1 ± 0.5	128 ± 2	128 ± 2	1 ± 0.5	2 ± 0.5
*C. glabrata* ORL20	131 ± 1	131 ± 1	2 ± 0.5	1 ± 0.5	128 ± 2	256 ± 2	2 ± 0.5	4 ± 0.5
*C. glabrata* ORL22	131 ± 1	131 ± 1	1 ± 0.5	1 ± 0.5	128 ± 2	256 ± 2	1 ± 0.5	4 ± 0.5
*C. glabrata* ORL13	131 ± 1	131 ± 1	2 ± 0.5	1 ± 0.5	128 ± 2	256 ± 2	2 ± 0.5	4 ± 0.5
*C. tropicalis* ORL18	4.1 ± 0.25	8.2 ± 0.5	0.5 ± 0.25	1 ± 0.5	1 ± 0.5	2 ± 0.5	0.03 ± 0.005	0.03 ± 0.005
*C. tropicalis* ORL19	66 ± 0.5	8.2 ± 0.5	1 ± 0.5	1 ± 0.5	1 ± 0.5	2 ± 0.5	0.03 ± 0.005	0.03 ± 0.005
*C. tropicalis* ORL20	4.1 ± 0.25	8.2 ± 0.5	0.5 ± 0.25	1 ± 0.5	1 ± 0.5	2 ± 0.5	0.03 ± 0.005	0.03 ± 0.005
*C. tropicalis* ORL21	4.1 ± 0.25	8.2 ± 0.5	1 ± 0.5	1 ± 0.5	1 ± 0.5	2 ± 0.5	0.03 ± 0.005	0.03 ± 0.005
*C. tropicalis* ORL22	4.1 ± 0.25	8.2 ± 0.5	0.5 ± 0.25	1 ± 0.5	1 ± 0.5	2 ± 0.5	0.03 ± 0.005	0.03 ± 0.005
*C. tropicalis* ORL23	4.1 ± 0.25	8.2 ± 0.5	0.5 ± 0.25	1 ± 0.5	2 ± 0.5	2 ± 0.5	0.03 ± 0.005	0.03 ± 0.005
*C. parapsilosis* ORL25	16.4 ± 0.5	20.5 ± 0.5	0.5 ± 0.25	1 ± 0.5	2 ± 0.5	4 ± 0.5	0.125 ± 0.05	0.250 ± 0.05
*C. parapsilosis* ORL25	16.4 ± 0.5	16.4 ± 0.5	1 ± 0.5	1 ± 0.5	2 ± 0.5	4 ± 0.5	0.125 ± 0.05	0.250 ± 0.05
*C. parapsilosis* ORL27	16.4 ± 0.5	20.5 ± 0.5	0.5 ± 0.25	1 ± 0.5	2 ± 0.5	4 ± 0.5	0.125 ± 0.05	0.250 ± 0.05
*C. parapsilosis* ORL28	16.4 ± 0.5	16.4 ± 0.5	0.5 ± 0.25	1 ± 0.5	2 ± 0.5	4 ± 0.5	0.125 ± 0.05	0.250 ± 0.05
*C. parapsilosis* ORL29	16.4 ± 0.5	16.4 ± 0.5	0.5 ± 0.25	1 ± 0.5	2 ± 0.5	4 ± 0.5	0.125 ± 0.05	0.250 ± 0.05
*C. parapsilosis* ORL30	16.4 ± 0.5	20.5 ± 0.5	0.5 ± 0.25	1 ± 0.5	2 ± 0.5	4 ± 0.5	0.125 ± 0.05	0.250 ± 0.05

**Table 3 jof-07-00383-t003:** Synergistic activity of REO in combination with fluconazole and amphotericin B.

Strains	MIC Alone (µg/mL)			MIC in Combination		FICI		Effect	
	REO	Flz	Amp B	REO-Flz	REO-Amp B	REO-Flz	REO-Amp B	REO-Flz	REO-Amp B
*C. albicans* ORL3	4.1	2	0.25	8.2–0.03	0.52–0.06	2.02	0.38	Ant	Syn
*C. albicans* ORL8	2.05	2	0.25	2.05–0.03	0.52–0.06	1.02	0.5	N.I	Syn
*C. tropicalis* ORL20	2.05	2	0.12	2.05–0.03	1.03–0.06	1.02	1.5	N.I	N.I
*C. tropicalis* ORL21	2.05	2	0.06	2.05–0.03	1.03–0.03	1.02	0.8	N.I	Syn

**Amp B**: amphotericin B; **Flz**: fluconazole; Ant: antagonistic activity; Syn: synergistic activity; N.I: no interaction.

**Table 4 jof-07-00383-t004:** Time for REO to reach 50%, 90%, and 99.9% growth reduction concerning start inoculum to *C. tropicalis* and *C. albicans* strains.

Strain	Growth Reduction	REO ^1^
*Candida tropicalis* ORL21	50%	0.39
	90%	1.57
	99.9%	1.79
*Candida tropicalis* ORL20	50%	0.88
	90%	1.65
	99.9%	2.93
*Candida albicans* ORL08	50%	4.51
	90%	N.A.
	99.9%	N.A.
*Candida albicans* ORL03	50%	3.63
	90%	N.A.
	99.9%	N.A.

N.A.: not achieved. ^1^ Treated at 37 °C for 48 h.

## Data Availability

The data presented in this study are available in the article or Appendix A.

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
