# Peer review of "Colombian Essential Oil of Ruta graveolens against Nosocomial Antifungal Resistant Candida Strains"

_jof, 2021, doi:10.3390/jof7050383_

Round 1
Reviewer 1 Report
Overall, a good paper,, although it contains some major language mistakes. As such, I recommend the manuscript to be copy proofed by a native speaker.
Concerning the antibiofilm activity, it is not clear how you selected the 4 isolates that you tested.
Also in line 231 All the strains tested produced biofilms on polystyrene microplates after 48 h of 231 incubation at 37 °C, as shown in Table 1. The table does not contain any reference to biofilm.
Furthermore, I strongly recommend that antibiofilm activity be compared with a standard antibiofilm agent. Any interpretation of results is void of significance if you do not provide a standard.
Author Response
Manuscript ID: JOF-1144553
Colombian essential oil of Ruta graveolens against Candida sp. isolated from the oral cavity of patients with head and neck cancer
The authors are very grateful to reviewer 1 of the manuscript for their valuable comments and suggestions. The manuscript authors' responses to the reviewer comments are provided below.
|
Comments from Reviewer |
Answers from authors |
|
Overall, a good paper, although it contains some major language mistakes. As such, I recommend the manuscript to be copy proofed by a native speaker.
|
Thank you for the evaluation and we are pleased that you are interested in the paper. The Paper was grammatically corrected by a mother tongue. |
|
Concerning the antibiofilm activity, it is not clear how you selected the 4 isolates that you tested.
|
R// We select the strains which showed high sensibility to the different treatments. In Lines 235-236 we specify “For the TKK assay two representative strains of C. albicans and C. tropicalis which showed high sensibility to REO were taken into consideration”. |
|
Also, in line 23. All the strains tested produced biofilms on polystyrene microplates after 48 h of incubation at 37 °C, as shown in Table 1. The table does not contain any reference to biofilm.
|
R// Sorry for the oversight, in line 273 in the text we specify “Figure 2” |
|
Furthermore, I strongly recommend that antibiofilm activity be compared with a standard antibiofilm agent. Any interpretation of results is void of significance if you do not provide a standard.
|
R// We appreciate the reviewer's suggestion, we tested the different strain with the Amphotericin B a well-known antibiofilm agent as reported by Mukherjee et al 2009. In lines 272-282 we specify that: “All the strains tested produced significant biofilm biomass on polystyrene micro-plates at 37 °C, as shown in Figure 2. The two C. tropicalis and C. albicans ORL3 strains formed a strong biofilm biomass, while C. albicans ORL08 showed a weak production. When 1% of REO was applied to the formed biofilm, we observed a significant detachment of the biofilm biomass after 1 hour of REO exposure, overall in C. albicans (41.2 and 36.1 % for ORL08 and ORL03 strains respectively). On the contrary, we observed a negligible biofilm detachment in samples with C. tropicalis during the first hour of treatment. Comparing the REO efficacy with that of amphotericin B, on the biofilm eradication, we observed a slight but not significant (p>0.05) differences between the two treatments. It is to worthy to mention that, the strongest biofilm formed by C. albicans ORL03 was more difficult to eradicate with both REO and amphotericin B.
In lines 407-409: The percentage of biofilm eradication here obtained were similar to those obtained with Amphotericin B, a drug with known efficacy against Candida biofilm [39].
In lines 423-425: Considering the above results, further studies will be address to increase the concentration and exposure time of REO to obtain the eradication of the C. tropicalis biofilm. |
Sincerely,
The authors

Reviewer 2 Report
The manuscript by Donadu et al. describes the antifungal properties of the Ruta graveolens essential oil against four species that belong to the Candida genus. The subject is interesting, but I regret to mention the study seems to be in a preliminary stage and requires additional work to be considered a solid piece of research of interest for the medical mycology community.
As a must to do, I consider that the authors should address the possible mechanisms of this oil when interacting with the fungal cells, does it affect the cell wall? the plasma membrane? the fungal metabolism? how does it perform such an effect? Moreover, the synergism and antagonisms with well-described antifungal drugs should be investigated. Does the oil show cytotoxicity in mammalian cells at the MFC?
As minor points:
Since the experiments were conducted at 37 °C, what was the final cell morphology?
How were the fungal strains identified? Please detail "standard methods"
How many batches of oil were used in the study?
What is the relevance of detail that the essential oil comes from Colombia? is it expected that the composition changes dramatically if prepared from plants from different countries?
What is the relevance of specifying that strains were isolated from the oral cavity from cancer patients? is it expected that the oil effect changes if fungal strains are isolated from other parts of the body or another type of patient?
The English usage must be improved, the text contains many grammar mistakes, awkward sentences, and typos. As an example, the title has a typo.
Author Response
Manuscript ID: JOF-1144553
Colombian essential oil of Ruta graveolens against Candida sp. isolated from the oral cavity of patients with head and neck cancer
The authors thank very much the Reviewer 2 of the manuscript for their valuable comments and suggestions. The responses from authors of the manuscript to the comments made by the Reviewer are given below.
|
Comments from Reviewer |
Answers from authors |
|
The manuscript by Donadu et al. describes the antifungal properties of the Ruta graveolens essential oil against four species that belong to the Candida genus. The subject is interesting, but I regret to mention the study seems to be in a preliminary stage and requires additional work to be considered a solid piece of research of interest for the medical mycology community.
|
R// We appreciate the reviewer's comment, following your suggestions we deepen in the possible mechanism of action of this oil by monitoring of nucleic acids, extracellular pH, and the membrane cell integrity of the yeasts, these measurements are an index of cell lysis and is generally used as an indicator of gross and irreversible damage to the cell membrane. Lines 170-191: 2.7.Leakage of DNA and RNA through the fungal membrane The release of cellular contents was determined according to the method by [26] with some modifications, we used a yeast load 5.0 ± 0.5 log CFU/mL for the analysis. To determine the concentration of the released constituents at 0, 30, 60 and 120 min of treatment, 50 mL of supernatant after of centrifugation was used to measure the absorbance at 260 nm with a bio photometer (Eppendorf 6131, Eppendorf, Hamburg, Germany). Control sample without REO, with REO, and with Fluconazole were tested. 2.8. Measurement of extracellular pH Candida extracellular pH treated with REO 1% was determined according the methodology reported by [26], we used a yeast load 5.0 ± 0.5 log CFU/mL for the analysis. After the centrifugation of the cells, these were washed three times and resuspended with sterilized double distilled water. The measurements of extracellular pH of the samples at 0, 30, 60, and 120 min were carried out using a pH-meter (Mettler-Toledo), as well a control sample without REO, and with Fluconazole were tested. 2.9. Effect of the REO on the membrane integrity. Cell membrane damage produced by the REO was evidenced using Evans blue staining according to Chaves-Lopez et al., [27]. 20 μl of the Candide suspension were incubated in coverslips in YPD broth at 37º C for 24 hours. Then, one of them for each strain were treated with 1% REO for one hour and they stained with Evans blue for five minutes. No treated samples were considered as control. Samples were observed under microscopy (Nikon ECLIPSE E 200, Nikon, Melville, NY). Photographs were taken with Samsung COLOR CAMERA SAC-410 PA interfaced with a PC.
In lines 294-303 The mechanism of the action the antifungal activity of Essential oils is not clearly, some authors suggest that causes significant membrane damage, due to the destruction of the membrane integrity [28]. The effect of REO on Candida membrane degradation and the release cellular constituents at 0, 30, 60, 90, and 120 min after the treatment were determined, and the results are shown in Figure 3. As observed in Figure 3 the exposure of C. tropicalis and C. albicans to REO, lead a significant (P < 0.05) increase in cellular release which was intensify with the exposure time. Indeed, an early release of the intracellular compounds was observed already after 30 min of exposure to REO, in comparison with the untreated sample. A minimal and constant cellular release in the Candida yeasts were observed with the fluconazole.
In lines 314-321: The extracellular pH of Candida cells exposed to 1.0% REO, fluconazole, and un-treated are presented in Figure 4. The extracellular pH decreased in the control. Regarding to the extracellular pH of the C. Tropicalis yeasts treated with REO showed a constant behavior in the initial 30 min after of treatment, followed by a significant increase (P < 0.05) of 15.2 and 14.2% after 120 min treatment for ORL21 and ORL20, respectively, concerning the control. C. albicansyeasts showed a minor but significant increase (P < 0.05) about control with 3.7 for ORL08, and 7.6% for ORL03. No significant differences were evidenced in the yeasts with the fluconazole after 120 min post treat-ment. In lines 330-339 To investigate if there was a disruption cell membrane integrity to the exposure to the REO, the cells were stained with Evans blue. As indicated in Figure 5, the results showed that when the yeast are treated with the REO and observed under a light microscope the majority of the cells were blue stained, suggesting that the cell membranes were compromised after 1 hour of treatment with the essential oil. Therefore, REO may act on the cellular membrane affecting its integrity, and consequently resulting in an increased intracellular leakage of the macromolecules, thus confirming that the plas-ma membrane is a target of the mode of action of REO against C. albicans and C. tropicalis
In lines 426-446. During recent years, the antifungal activity of the REO have been reported (16-20); however, their mode of action, is not yet clearly understood. Thus, in the pre-sent study, we used different approaches to provide insights into the antifungal activ-ity of REO. To this purpose we evaluated the release cellular, the extracellular pH and the cell membrane integrity in Candida. Some authors have suggested that the antimi-crobial activity of essential oils involves phenomena such as changes of cell membrane integrity, leading to an alteration of permeability and a consequent leakage of cell contents [45]. In our experiment we observed a markedly release of cell constituents during the time with a notably icell staining with the blue Evans, indicating that the permeability of the cell membrane was compromised. In addition, REO clearly induced the leakage of intracellular protons as evidenced by the increase in extracellular pH. Our results suggest that the REO accumulation in the cell membrane can induce loss of integrity changing homeostatic environment, allowing ions leak and cellular release, which could be responsible for the antifungal activity of this essential oil. No signifi-cant cellular release or changes in extracellular pH were presented with fluconazole evidencing the different mechanism of action that presents the azoles [46]. Some au-thors have reported similar results with others essential oils, Chen et al., [45] reported that Anethum graveolens essential oil induced a lesion of the cell membrane in C. albi-cans. On other hand, Ahmad et al., [47] reported that mechanism of action of the Cori-aria nepalensis essential oils disrupt of membrane integrity in different Candida isolates. A similar result was reported by Rajkowska et al., [48] using different essential oils against Candida.
In addition, we added the Figures 3, 4, 5.
|
|
Since the experiments were conducted at 37 °C, what was the final cell morphology?
|
R// In lines 338-339 we specify that: In addition, we observed a cell shrinkage after the treatment with REO probably due to the release of the intracellular components. |
|
How were the fungal strains identified? Please detail "standard methods"
|
The strains were identified previously. In the text we specify in lines 91-93 as follow: All microorganisms were identified by standard methods: germe tube test and stored on Sabouraud dextrose agar plates until the study was performed [22].” |
|
How many batches of oil were used in the study? |
In line 97-98: Rue essential oil (REO) was obtained from Kräuter SAS (Bogotá- Colombia) lot n ° SSTE01
|
|
What is the relevance of detail that the essential oil comes from Colombia? is it expected that the composition changes dramatically if prepared from plants from different countries? |
R// It is well known that the qualitative and quantitative variability of the essential oils depend on several factors such as genetic pool, stage of plant growth, environmental and geographical conditions including soil composition, climatic conditions, seasonal variations, soils characteristics, part of the plant and methods of extraction. The composition of our essential oil is similar to that the other countries regarding the compound profile, but differs on their quantities, overall in the minor components. In this regard, several studies suggest that some major antimicrobial constituents combined with other minor constituents might be involved in improving the overall antimicrobial activity of the oil. For this reason, we considered important to mention the origin of the oil. |
|
What is the relevance of specifying that strains were isolated from the oral cavity from cancer patients? is it expected that the oil effect changes if fungal strains are isolated from other parts of the body or another type of patient? |
We thank the Reviewer for this consideration. R//Many Factors contribute to clinical Candida antifungal drug resistance among of them Immune status, Site of infection, Severity of infection, Presence of foreign materials (dentures, catheters, prosthetic valves). Moreover, some studies have shown that C. albicans isolates from different clinical samples produce one or more virulence factors with a greater or lesser frequency depending on the place of isolation, the clinical condition of the patient, their immune status and the therapeutic regimen. (M.P. Giolo, T.I.E. Svidzinski. Physiopathogenesis, epidemiology and laboratory diagnosis of candidemia. Jornal Brasileiro de Patologia e Medicina Laboratorial, 46 (3) (2010), pp. 225-234). As we don’t study different Candida species from different sites of isolation, we emphasize that they were isolated the oral cavity of patients with head and neck cancer. In addition, there is currently growing concern about the relationship between microbial infections and cancer. More and more studies support the view that there is an association, especially when the causative agents are fungi. Ramirez-Garcia et al. evidences that the opportunistic fungus Candida albicans increases the risk of carcinogenesis of the oral cavity and of the neck (being more treated with chemotherapy and radiotherapy) and metastasis. Recent publications show that C. albicans is able to promote cancer through several mechanisms: production of carcinogenic by-products, activation of inflammation, induction of the Th17 response and molecular mimicry. We emphasize the need not only to control this type of infection during cancer treatment with alternative drugs such as molecules of natural origin and used in folk medicine, especially given the important role of this yeast species in nosocomial infections.
|
|
The English usage must be improved, the text contains many grammar mistakes, awkward sentences, and typos. |
The Paper on her welcome advice was grammatically corrected by a mother tongue. |
Sincerely,
Prof. Clemencia Chaves Lopez and Dr. Francesca Maggio
with co-authors

Reviewer 3 Report
The manuscript submitted by Donadu et al .describes the antifungal activity of Ruta graveolens using various in vitro characterization methods. The methodologies used are up-to-date, the results and significant, novel and an important addition to the scientific literature. Overall, the paper is relevant to the journals audience and should be accepted, pending some minor remarks (see below) that should be addressed before the paper is processed further:
General (abstract and main text):
Please adhere to the international guidelines for writing microorganism names (or refer to the paper for istructions: https://www.aph-hsps.hu/acta/index.php/aph/article/view/15), i.e. first mention: full name (e.g. Candida albicans), any subsequent mention of the same genus, abbreviated name: C. albicans, C. parapsilosis, and always in italics! The same goes for R. graveolens.
please do not capitalize the name of antifungal drugs
Abstract:
please rephase the first two sentences into one concise sentence
In the last years, … reducing the levels of antifungal resistance
Candida albicans, C. parapsilosis, C. glabrata, C. tropicalis., in the following, use C. albicans and C. tropicalis
time-kill kinetics assay
fungicidal effect
Keywords: please consider to use the following keywords: Candida spp., head and neck cancer
- Introduction:
please rephase the first two sentences into one concise sentence. In addition, please consider detailing that Candida spp. may also affect other mucosal surfaces. Consider introducing the following reference:
https://www.ncbi.nlm.nih.gov/pmc/articles/PMC6715075/
the more commonly used abbreviation is NAC for non-albicans Candida
…widely within the last ten years.
Please write 1-2 sentences on azole drugs being the backbone of antifungal therapy.
Please briefly discuss the difficulty of marketing novel antimicrobial drugs and then introduce essential oils as a potential alternative strategy to combat various infections. Use the following reference:
https://www.ncbi.nlm.nih.gov/pmc/articles/PMC6429336/
In order to find new classes of antifungals...
L53-55: please rephrase this sentence to be more concise
L57-58: Please remove this sentence
L58-L63: please rephase this sentence to be more concise
In this regard, previous studies have demonstrated
Ruta graveloens on first mention and then R. graveolens
L67-70: note the correct writing of bacterial names!
Methods:
L86-87: please provide some more details on the identification methods used during the ID of the fungal isolates
L91: 0.14 V/V%?
L103: Please include the following reference:
https://pubmed.ncbi.nlm.nih.gov/31771095/
2.5. Time-kill kinetics assay
Chavez-López et al. with some modifications
In order to determine the fungistatic or fungicidal character of REO
To assess biofilm-formation, 200 uL…
Results:
please insert a space between numbers and ug/mL values
0.005-16 V/V%
synthetic drugs, such as…
what was the interpretation criteria based on for proclaiming resistance to general antifungals? CLSI or EUCAST?
L225: this sentence needs to be rephased and corrected.
L238: this sentence needs to be rephased and corrected.
Discussion:
L244: Antifungal drug resistance may be due to…
L282: The results suggest that the REO effect is species-specific against Candida.
I suggest that the authors include a separate Conclusions section, and move and complement the last paragraph of the text to it.
Author Response
Dear reviewer 3
we send you the manuscript attached, with the corrections you requested and we thank you for your valuable comments.
The linguistic correction was carried out by a native English speaker from the University of Sassari.

Round 2
Reviewer 1 Report
I consider that the improvements provided by the authors meet my initial recommendations.
Author Response
Dear Reviewer
We thank you for the valuable advice.
The authors
Reviewer 2 Report
The manuscript has been significantly improved, and I thank the authors for the modifications. However, there are still points that the authors have to address to have a study up to the publication level.
- Nowadays, strains identification to the species level should be confirmed by molecular means, which is not the case in this study. I encourage the authors to provide this relevant information.
- The synergistic/antagonistic combination of this oil with conventional antifungal drugs was suggested in my previous assessment and not addressed by the authors in the rebuttal letter or the modified version of the manuscript. Therefore, my previous comment stands.
- I insist again, in my view, there is no point to specify the origin of clinical isolates in the manuscript title, it provides no relevant information and makes the title hard to read.
- The English usage was improved but the manuscript still contains typos and grammar mistakes.
Author Response
Manuscript ID: JOF-1144553
Colombian essential oil of Ruta graveolens against nosocomial antifungal resistant Candida strains
The authors thank very much the Reviewer 2 of the manuscript for their valuable comments and suggestions that helped us to improve our paper. The responses to the comments made by the Reviewer are given below.
|
Comments from Reviewer |
Answers from authors |
|
Nowadays, strains identification to the species level should be confirmed by molecular means, which is not the case in this study. I encourage the authors to provide this relevant information.
|
R// Thanks for your suggestion. The identification of all the strains (24 strains) by molecular methods is part of other further work. Considering that the germ tube test (GTT) and YBC Vitek Card (Bio-Merieux) are well known standard methods to differentiation of Candida albicans from non-albicans, toconferm the identity of C. tropicalis, we decide to identify by molecular method only the strains of this specie that were choise to perform all the experiment here reported. In the manuscript we reported in lines 97-100 the Gen Bank accession number of the ITS sequences obtained by amplification of the ITS region (ITS1-5,8S-ITS2) with universal fungal primers (ITS1, ITS4). |
|
The synergistic/antagonistic combination of this oil with conventional antifungal drugs was suggested in my previous assessment and not addressed by the authors in the rebuttal letter or the modified version of the manuscript. Therefore, my previous comment stands. |
R// We appreciate the reviewer's comment, the synergistic potential of Ruta graveolens essential oil in combinations with fluconazole anda amphotericin B was evaluated. In order to uniform the measurement units we converted the percentage (%) of the REO in (µg/ml).
In lines 139-160 we included the methodology used by evaluation of the synergistic potential of the combinations Ruta graveolens essential oil and two antifungal drugs. “2.5. Synergistic potential of REO with antifungal antibiotics against C. albicans and C.tropicalis In order to determinate the synergy between antifungal antibiotics and Ruta graveolens essential oil, the checkerboard method was performed to obtain the Fractional inhibitory concentration indices (FICI) of REO in combination with Amphotericin B and Fluconazole following the methodology proposed by [24]. The microtiter plates were filled with a combination of 50 μl REO and 50 μl antibiotics (Fluconazole and Amphotericin B) at MIC diluted twofold in a serial manner in YPD broth with a 0,01% of 2,3,5-Triphenyltetrazolium chloride (TTC). Successively, 10 μl of fungal culture were seeded (106 CFU / mL) in each well and mixed well. The plates were incubated at 37 °C for 48 h. The inhibition of growth of fungal cells was indicated by absence of the red color.
The FICI were calculated using the following formulas:
The synergistic potential was evaluated with the novel interpretation of Fratini et al., [25]: synergistic effect (FIC Index ≤1.0), commutative effect (FIC Index= 1); no interaction (FIC Index >1.0–≤2.0) and antagonistic effect (FIC Index >2.0).”
In lines 261-276. “To overcome the mechanisms of bacteria resistance against antibiotics, in the last years, some studies have proposed the use of association of plant extracts with antibiotics. In our study the synergistic potential of essential oil of Ruta graveolens in combination with fluconazole and amphotericin B using checkerboard method was evaluated. As shown in Table 3, REO showed synergistic effects with amphotericin B against C. albicans ORL3 and ORL8 and C. tropicalis ORL21 with FICI value of 0.38, 0.5 and 0.8, respectively. REO did not present interaction with the Amphotericin B in C. tropicalis ORL20. Regarding the combination REO-fluconazole, no interactions were evidenced with C. tropicalis or C. albicans ORL8. However, antagonistic activity was revealed in C. albicans ORL3. As is observed in Table 3, the concentrations used to achieve the synergistic activity of the combinations were considerably lower than those of the MIC of oil and antibiotics used and despite in some combinations no evidenced synergy if a decrease in the individual concentration used in the drugs was reached.”
In addition, we added the Table 3.
In lines 422-437. “Antibiotic resistance is a big concern around the world and many strategies have been adopted to reduce this problem. In the last years, a rational approach to deal with antibiotic resistance problems using a combination therapy combining conventional antibiotics and essential oils, have been proposed. In this context, the interaction between plant extracts and antibiotics with synergistic activity against Candida species has been reported. For example, Saad et al., [39] reported that geraniol displayed synergistic effect with both fluconazole and amphotericin B. Also, the synergistic effect of thymol and nystatin was observed on Candidaspecies [40]. In the same way, Citrus aurantium essential oil showed synergistic potential with fluconazole and amphotericin B against C. albicans and enhanced the antifungal efficacy of the clinical drugs by 8.3 to 34.4 folds [24]. In this study, we reported for the first time a synergistic effect of REO with amphotericin B against C. albicans and C. tropicalis. Combination of clinical antibiotics with essential oils and phytocompounds targeting resistant fungi may have different mechanism of action including i) sequential inhibition of common biochemical pathways, ii) amplified diffusion of one antifungal agent subsequent from the action of another antifungal agent on the fungal cell membrane; iii) inhibition of different targets and iv the inhibition of carrier proteins [40]”
In lines 503-505 In conclusion section we include: Fractional Inhibitory Concentration Index (FICI), showed synergic effects of REO combined with amphotericin B. These results suggest the possible effective use of REO alone or in combination with this antibiotic, against multi-drug resistant overall on C. tropicalis strains
|
|
I insist again, in my view, there is no point to specify the origin of clinical isolates in the manuscript title, it provides no relevant information and makes the title hard to read.
|
R// We appreciate the reviewer's comment, the title was changed in: “Colombian essential oil of Ruta graveolensagainst nosocomial antifungal resistant Candida strains”. |
|
The English usage was improved but the manuscript still contains typos and grammar mistakes. |
The Paper on her welcome advice was grammatically corrected by a mother tongue. |
Sincerely,
The authors
